# Cosmic Covers: A Geometry-First Pipeline for 3D Comoving Astronomical Catalogs

János Horváth
Visionary Tech & Event Solutions
USA, California, Sacramento

horvath_janos@visionarytecheventsolutions.com

István Horváth
University of Public Service
Budapest, Ludovika tér 2.

horvath.istvan@uni-nke.hu

## Abstract

*We present a fully reproducible pipeline for turning heterogeneous public astronomical catalogs into a unified* 3D comoving point-cloud *representation and for summarizing these distributions with* geometric covers. *Starting from the ZTF Bright Transient Survey (BTS) supernova sample, the Swift/BAT GRB catalog (with redshifts), and the redMaPPer cluster catalogs, we (i) harmonize metadata and categories across surveys, (ii) convert sky coordinates and redshift to Cartesian comoving coordinates using a standard cosmology, (iii) fit density baselines (Gaussian kernel density estimation and noise-contrastive density ratio models), and (iv) build compact cover sets (K-means and K-center) that approximate the spatial support and yield interpretable "representative" events with error radii. The same codebase generates publication-ready summary tables and a large suite of plots, including sky footprints, distance distributions, and cover allocation visualizations (static and interactive). On the processed releases used in this workshop paper, the pipeline produces $N = 11514$ SNe, $N = 522$ GRBs and $N = 26898$ clusters; KDE baselines and density-ratio AUCs provide a reference point for future method development.*

## 1. Introduction

Modern time-domain surveys and mission archives provide rich catalogs of transient and variable phenomena. Yet, the underlying data sources are heterogeneous: event names and schema differ across archives; sky coordinates are reported in different conventions; redshift measurements are incomplete and often selection-biased; and many astrophysical categories (e.g., supernova subtypes) are noisy or ambiguous. These issues make it difficult to build *geometric* or *machine-learning* baselines that are simultaneously reproducible, interpretable, and suitable for workshop-scale experimentation.

This paper focuses on a simple but useful abstraction: represent each event by its position in *3D comoving space*, and then study the resulting point clouds with density models and geometric summarization. In astronomy, converting $(\alpha, \delta, z)$ (right ascension, declination, redshift) to comoving Cartesian coordinates is standard; in ML, representing data as point clouds enables a large toolbox of methods (clustering, coresets, covering numbers, neighborhood graphs, density estimation, and representation learning). Our goal is not to claim a new astrophysical discovery, but to provide a *strong, end-to-end baseline* that workshop participants can build upon.

**Contributions.**
- **Unified data ingestion.** We download and validate a curated set of public endpoints for ZTF BTS SNe, Swift/BAT GRBs, redMaPPer clusters, and AstroCats quantities, and we separate outputs by dataset and category.
- **Geometry-first representation.** We map each catalog to 3D comoving coordinates (and optional log-distance coordinates) with transparent filtering and provenance, using `astropy` cosmology utilities [10–12].
- **Baselines for densities and covers.** We implement KDE likelihood baselines and noise-contrastive density-ratio tests [27], and we produce K-means and K-center cover sets [22, 38, 40] that summarize distributions with explicit error radii.
- **Visualization suite.** We generate a workshop-ready set of figures (static and interactive), including sky footprints, distance histograms, embedding plots, and cover allocation diagrams.

## 2. Related Work

**GRB catalogs, classification, and redshift provenance.** Catalog-level studies of gamma-ray bursts (GRBs) illustrate the same challenges that motivate our selection-aware null models: heterogeneous discovery pipelines,

evolving calibrations, and incomplete or inconsistently sourced redshifts. Instruments and catalogs such as Swift/BAT and Fermi/GBM remain the backbone for prompt-emission properties and redshift aggregation [21, 37, 41, 43, 57], but duration/hardness-based class labels are known to be ambiguous near decision boundaries [35]. This ambiguity is highlighted by the classification of GRB 170817A/GW170817 in the Fermi duration–hardness plane [31], and by evidence for an intermediate-duration population in the Swift era when restricting to bursts with measured redshifts and consistent follow-up provenance [15]. Pulse-level measurements provide complementary structure beyond $T_{90}$, e.g., via BATSE TTE-derived pulse catalogs for short GRBs [28], while high-redshift events such as GRB 080913 stress-test classification and energetics under cosmology- and selection-dependent transformations [48]. Looking forward, mission concepts emphasizing broadband spectroscopy and polarimetry (and thus more direct constraints on spectral features and environments) motivate why future catalogs may shift which quantities are reliably available at scale [24]. Across these regimes, we treat names/identifiers and redshift provenance as first-class metadata, leveraging community infrastructure for rapid alerts and curation [1, 5], and we link GRB-side metadata to optical time-domain discovery streams and curated transient bookkeeping (e.g., ZTF/BTS and its automation) to keep selection effects explicit in downstream analyses [6, 9, 14, 16, 23, 26, 33, 52, 59, 63]. Finally, because redshift interpretation can be sensitive to model assumptions and historical systematics debates, we view cosmology choices and redshift handling as part of the reproducibility surface of any cross-catalog population claim [42].

**Time-domain surveys, catalogs, redshifts, and provenance.** Modern time-domain surveys produce heterogeneous transient samples with strong selection effects, irregular cadence, and incomplete redshift information. Our transient sample is anchored in the Zwicky Transient Facility (ZTF) alert stream and survey operations [6, 23], and leverages the ZTF Bright Transient Survey (BTS) ecosystem, including automated triage and bookkeeping at BTS scale (e.g., BTSbot) [52]. For Type Ia supernovae, large homogeneous BTS releases (e.g., BTS Type Ia DR2) provide curated metadata and updated calibrations useful for benchmarking and population studies [16]. More broadly, time-domain discovery is driven by wide-field surveys such as Pan-STARRS1 [9], ATLAS [63], and ASAS-SN [59], with forthcoming LSST-scale alert streams [33]. We treat catalog provenance and identifiers as first-class metadata, consistent with community infrastructure for naming and rapid alerts such as the Transient Name Server (TNS) [1] and the GRB Coordinates Network (GCN) [5]. GRB properties and redshifts are obtained from the Swift/BAT catalogs [37, 57], with complementary population context from Fermi/GBM [21, 41, 43] and the classical duration-based taxonomy [35]; we also note complementary optical/afterglow catalog efforts emphasizing consistent redshift provenance and multiband evolution [14]. For supernova metadata, we support importing auxiliary quantities from community catalogs via programmatic interfaces such as the Open Astronomy Catalogs / AstroCats API [26]; in the specific run reported here, the queried quantity endpoints returned empty/placeholder tables and are treated as missing data.

**Galaxy-cluster catalogs and selection functions.** For large-scale structure context, we draw galaxy clusters from redMaPPer, a red-sequence finder with a calibrated richness proxy and well-studied selection functions [56]. To contextualize sky coverage and redshift reach, we ingest auxiliary redMaPPer footprint products and note complementary regimes from SZ-selected catalogs (*Planck* [50], SPT [7], ACT [30]) and optical SDSS compilations such as WHL [65]. These references motivate our emphasis on selection-preserving null models and diagnostics that separate footprint effects from intrinsic 3D structure.

**Geometry-first summarization, coresets, and coverings.** A recurring challenge in time-domain astronomy is that decision-making and visualization must often occur under resource constraints (limited follow-up time, heterogeneous instruments, and incomplete labels). Geometric summarization offers a complementary lens: rather than committing early to a parametric population model, one can compute covers or coresets that preserve salient structure for downstream tasks such as representative target selection, interactive inspection, or weakly supervised labeling. Classic objectives include the $k$-center (minimax) problem, which admits a simple farthest-first (Gonzalez) approximation [22], and $k$-means clustering [38, 40], with practical seeding strategies such as $k$-means++ [2]. Coreset constructions give principled small summaries for clustering objectives [4, 20] and also appear in active learning as coreset selection [58]. Recent work extends these ideas toward settings closer to real pipelines, including robustness to deletions and dynamic/streaming updates; for example, deletion-robust coresets for $k$-center help maintain covers under iterative catalog revisions [36], and near-optimal approximation guarantees with explicit multiplicative/additive error control strengthen the link between geometric covers and error budgets [13]. Related sampling primitives are common in point-cloud learning (e.g., farthest point sampling in PointNet++ [51]) and in geometric deep learning more generally [8]. In our pipeline, we use covers to summarize catalog geometry, select representative targets, and visualize "coverage" as a function of the number of centers.

**Point-cloud representation learning and modern 3D models.** Beyond purely geometric objectives, modern point-cloud representation learning provides powerful learned embeddings for point sets. Point Transformer introduces attention-style operators directly on point clouds [67]. Masked pretraining strategies for 3D data, inspired by masked autoencoders [29], have been adapted to point clouds to learn transferable encoders from unlabeled geometry; Point-BERT and Point-MAE are representative examples [44, 66]. Although our pipeline emphasizes interpretable geometric covers (to keep error semantics transparent and reduce dependence on large labeled training sets), these learned representations are a natural extension: embeddings can replace raw $(x, y, z)$ coordinates or augment them with measurement-aware features, potentially improving downstream clustering, anomaly detection, or domain adaptation [8].

**Density estimation, density ratios, and two-sample testing.** Kernel density estimation (KDE) is a standard baseline [47, 55] but can be sensitive to bandwidth in sparse, high-dimensional settings. Modern alternatives include flow-based density models [18, 19, 45, 46, 53] and score-based methods [32, 60], yet scientific applications often benefit from diagnostics that do not require a fully normalized likelihood. Noise-contrastive estimation (NCE) [27] yields a flexible density-ratio proxy, and related two-sample tests based on classifiers [39] or kernels (MMD) [25] provide complementary views; we also note optimal-transport distances as geometry-aware comparisons between empirical distributions [49, 64]. In parallel, high-dimensional generative modeling has advanced rapidly: score-based generative modeling via SDEs provides a unifying view of diffusion/score matching [61], diffusion models show strong empirical performance [17] with further clarification of design and numerical behavior [34], latent diffusion improves scalability [54], and consistency models offer an alternative formulation aimed at reducing sampling cost [62]. In our pipeline, NCE quantifies how distinguishable a catalog is from a *selection-preserving null* (Section 5), providing a conservative baseline for spatial "non-randomness".

**Cosmology, coordinates, and reproducible scientific software.** Converting heterogeneous sky coordinates and redshifts into physically meaningful 3D point clouds requires consistent cosmological assumptions and unit handling. We rely on standard community software (Astropy) for cosmology, coordinate transforms, and units, improving reproducibility and reducing implementation divergence across experiments [3].

Table 1. Download validation summary produced by the dataset fetch script.

| Metric | Value |
|---|---|
| Files scanned | 110 |
| Files OK | 110 |
| Files flagged | 0 |
| Total size [MB] | 129.9 |

| Catalog | $N$ | $z_{min}$ | $z_{med}$ | $z_{max}$ |
|---|---|---|---|---|
| SNe | 11514 | 0.0002 | 0.054 | 0.519 |
| GRB | 522 | 0.0133 | 1.7245 | 9.38 |
| Clusters | 26898 | 0.0811 | 0.3747 | 0.8943 |

Table 2. Summary of the observational catalogs used in our experiments after filtering to rows with valid sky coordinates and (when available) redshift.

## 3. Data and Preprocessing

We download all data sources listed in the accompanying scripts and validate files (non-empty, expected type). Table 1 summarizes the download status for the run associated with this paper.

### 3.1. ZTF BTS supernovae

We use multiple BTS explorer endpoints (default sample, CANTRANS subsample, "all" subsample, and quality filters). These are merged into a single SNe table with a dataset provenance column. We keep entries with finite sky coordinates and (when available) redshift. We also derive a coarse supernova type label by mapping the BTS claimed type to a small set of families (Ia, II, Ibc, IIn, Other).

### 3.2. Swift/BAT gamma-ray bursts

We parse the Swift/BAT "summary general" tables and join them with redshift lists when possible. We retain GRBs with finite $(\alpha, \delta)$ and finite redshift for the 3D representation. We additionally compute a simple duration label (short vs. long bursts) using the conventional T90 threshold at 2s.

### 3.3. redMaPPer galaxy clusters

We load the redMaPPer catalogs (DR8 and SVA1) and extract sky coordinates, richness, and redshift. Clusters are treated as a third point cloud and also serve as a reference population for transient–environment association tests.

### 3.4. Summary statistics

Table ?? reports dataset sizes and redshift ranges after preprocessing.

## 4. 3D Geometry Representation

**Spherical observables and reference frame.** Each object is represented by its sky position and redshift: right ascension $\alpha$ (deg), declination $\delta$ (deg), and redshift $z$. We treat the catalog coordinates as ICRS-equatorial coordinates as reported by the upstream sources. In practice, the pipeline accepts either decimal degrees or sexagesimal strings (e.g., `HH:MM:SS` for $\alpha$ and `DD:MM:SS` for $\delta$), and converts them into a common floating-point degree representation using `astropy.coordinates` [10, 11]. This ensures consistent handling of angle wrapping ($\alpha \in [0, 360)$), sign conventions for $\delta$, and heterogeneous formatting across catalogs.

**Cosmology and distance mapping.** To embed the catalog in three dimensions, we map redshift to a line-of-sight comoving distance. Assuming a flat $\Lambda$CDM cosmology with Planck18 parameters by default [12], the comoving distance is

$$d_c(z) = \frac{c}{H_0} \int_0^z \frac{dz'}{E(z')}, \qquad E(z) = \sqrt{\Omega_m (1+z)^3 + \Omega_\Lambda}, \tag{1}$$

where $c$ is the speed of light. We compute $d_c$ numerically and store it in comoving Mpc [10, 11]. For low-redshift events, peculiar velocities and redshift-definition choices (heliocentric vs. CMB frame) can introduce non-negligible distance uncertainty; because our goal is a robust geometric baseline rather than high-precision cosmography, we treat the reported $z$ as cosmological redshift and do not attempt additional flow corrections.

**Cartesian embedding in comoving Mpc.** We embed objects in a right-handed Cartesian coordinate system with the observer at the origin. Let $\alpha_r = \alpha \, \pi / 180$ and $\delta_r = \delta \, \pi / 180$ denote angles in radians. The unit direction vector is

$$\hat{\mathbf{u}}(\alpha, \delta) = [\cos \delta_r \cos \alpha_r, \; \cos \delta_r \sin \alpha_r, \; \sin \delta_r]^\top. \tag{2}$$

The comoving 3D position is then

$$\mathbf{x} = d_c(z) \, \hat{\mathbf{u}}(\alpha, \delta), \qquad \mathbf{x} = [x, y, z]^\top \in \mathbb{R}^3, \tag{3}$$

which yields the component-wise expressions

$$\begin{aligned} x &= d_c \cos \delta_r \cos \alpha_r, \\ y &= d_c \cos \delta_r \sin \alpha_r, \\ z &= d_c \sin \delta_r. \end{aligned} \tag{4}$$

This embedding preserves angular relationships locally and converts redshift ordering into a physically interpretable radial scale. It also makes Euclidean operations (nearest-neighbor search, covers, kernels, and neural embeddings) directly applicable: unless otherwise noted, distances in our geometric modules are standard Euclidean distances in comoving space, $d(\mathbf{x}_i, \mathbf{x}_j) = \|\mathbf{x}_i - \mathbf{x}_j\|_2$.

**Auxiliary radial coordinates and dynamic-range control.** Transient and cluster catalogs span a wide radial range; visualizations and certain cover constructions can become dominated by a small number of very distant objects. To improve numerical stability and perceptual interpretability, we store additional one-dimensional radial summaries:

$$r = d_c(z) \; (\text{Mpc}), \qquad \ell = \log_{10}\left(\frac{d_c(z)}{Mpc}\right), \tag{5}$$

and (optionally) the "hybrid" 3D coordinate $\tilde{\mathbf{x}} = [x, y, \ell]$ for covers and embeddings that benefit from a compressed radial axis. We emphasize that all scientific evaluation metrics (e.g., projected cluster association and 3D density tests) are performed in true comoving Mpc unless explicitly stated.

**Projected distances for association tests.** When comparing transients to galaxy clusters we separate angular and radial components. Given an event and a cluster at $(\alpha, \delta, z)$ and $(\alpha_c, \delta_c, z_c)$, we compute the angular separation $\theta$ on the sphere (haversine on $(\alpha_r, \delta_r)$), and convert it into a projected comoving-scale separation using the angular-diameter distance $D_A(z_c)$:

$$r_{\text{proj}} \approx \theta \, D_A(z_c). \tag{6}$$

We use this projected distance together with a redshift-window constraint $|z - z_c| < \Delta z$ to define a conservative "associated" label used in downstream diagnostic plots and AUC-based evaluations.

**Quality control and reproducibility.** Finally, we apply lightweight validation to guarantee a well-posed geometric representation: we remove rows with missing or non-finite $(\alpha, \delta, z)$, enforce plausible bounds (e.g., $\delta \in [-90, 90]$), and clip redshift to task-relevant intervals for each experiment (e.g., overlapping redshift ranges for transients and redMaPPer clusters). All intermediate columns (raw angles, parsed degrees, $d_c$, $\ell$, and Cartesian coordinates) are saved alongside provenance fields so that downstream experiments and figures can be regenerated deterministically from the processed CSVs.

## 5. Selection-Preserving Null Model

A key difficulty in interpreting 3D catalogs is that both sky position and redshift are strongly shaped by survey strategy, targeting cadence, host-followup policies, and measurement completeness. As a result, apparent "structure" in $(\alpha, \delta, z)$ space can arise even when the underlying astrophysical distribution is smooth, simply because the catalog is an incomplete, selection-biased sample of the sky. To provide a conservative baseline that directly targets this issue, we implement a *selection-preserving null* that keeps the observed

angular footprint fixed while destroying genuine 3D correlations.

**Construction.** Given a catalog of $N$ events with $(\alpha_i, \delta_i, z_i)$, we first convert each redshift to comoving distance $d_{c,i} = d_c(z_i)$ using the same cosmology as the main pipeline (Section 4). We then draw a random permutation $\pi$ of $\{1, \ldots, N\}$ and define a null catalog by

$$(\alpha_i, \delta_i, d_{c,i}^{\text{null}}) = (\alpha_i, \delta_i, d_{c,\pi(i)}). \qquad (7)$$

Finally, we recompute Cartesian coordinates for the null catalog using the original $(\alpha_i, \delta_i)$ but the permuted $d_{c,i}^{\text{null}}$. In words, each event keeps its sky position but "borrows" a distance from a different event.

**What it preserves (and what it breaks).** This null preserves: (i) the empirical sky coverage (including sharp footprint edges and masking patterns), (ii) the empirical one-dimensional distance/redshift distribution, and (iii) any catalog-specific angular inhomogeneities due to observing strategy. At the same time, it breaks any coupling between sky location and distance, which is the minimal requirement for removing true 3D correlations. Thus, if a method *cannot* distinguish the real catalog from this null, then any claimed "structure" should be treated cautiously: it may be indistinguishable from selection effects alone.

**Null ensembles and stability.** Because any single permutation yields only one null realization, we typically generate multiple null catalogs by using different random seeds (or different permutations) and report the distribution of the resulting test statistics (e.g., AUC or log-likelihood gaps). This enables uncertainty estimates for "detectability beyond selection" and reduces the risk that conclusions depend on a particular shuffle. In practice, we keep the null generation deterministic given a seed to make the pipeline fully reproducible.

**Why this is conservative.** The proposed null does *not* attempt to model the true selection function, which would require instrument- and survey-specific modeling. Instead, it asks a simpler, more defensible question: *is there evidence of 3D structure beyond what is already implied by the observed sky footprint and the observed distance distribution?* This is especially appropriate when combining heterogeneous catalogs (SNe, GRBs, clusters) whose selection functions differ substantially.

# 6. Baselines: Density Models

We evaluate two complementary baselines for quantifying spatial structure in comoving 3D: a simple likelihood-based density model and a discriminative density-ratio test against the selection-preserving null. Both baselines operate on Cartesian coordinates $\mathbf{p}_i = (x_i, y_i, z_i) \in \mathbb{R}^3$ in comoving Mpc (Section 4).

## 6.1. Gaussian KDE

Given points $\{\mathbf{p}_i\}_{i=1}^N \subset \mathbb{R}^3$, Gaussian KDE estimates a smooth density

$$\hat{p}(\mathbf{p}) = \frac{1}{Nh^3} \sum_{i=1}^N \exp\left(-\frac{\|\mathbf{p} - \mathbf{p}_i\|^2}{2h^2}\right), \qquad (8)$$

where $h$ is the bandwidth (in Mpc). We select $h$ from a fixed grid by maximizing the average log-likelihood on a held-out test split, and we report the average held-out log-likelihood as the KDE baseline score:

$$\text{LL}_{\text{test}} = \frac{1}{|\mathcal{T}|} \sum_{j \in \mathcal{T}} \log \hat{p}(\mathbf{p}_j). \qquad (9)$$

This metric is easy to compute and interpret as "how well a smooth local density model explains held-out points"; however, it is sensitive to coordinate scaling and to the curse of dimensionality, and it does not explicitly control for selection effects. For large catalogs, KDE also implicitly reflects survey footprint density variations, which motivates the null-based test below.

**Practical notes.** We use the same preprocessing and distance conversion for all catalogs to keep coordinate scales comparable. We also retain the comoving radial distance $d_c$ and optionally $\log_{10} d_c$ for downstream visualization and cover construction, but KDE is evaluated in $(x, y, z)$ unless otherwise stated.

## 6.2. Noise-contrastive density ratio

To measure structure *beyond* selection effects, we compare the real catalog $p(\mathbf{p})$ to a null distribution $q(\mathbf{p})$ induced by the selection-preserving shuffle in Section 5. Rather than fitting two normalized densities, we train a binary classifier to distinguish real samples $(y = 1)$ from null samples $(y = 0)$ using the logistic loss, which is a practical NCE-style setup [27]. Let $f_\theta(\mathbf{p})$ denote the classifier logit. Under standard assumptions, $f_\theta$ approximates a monotone transform of the log density ratio $\log \frac{p(\mathbf{p})}{q(\mathbf{p})}$, meaning that high scores indicate regions where the real catalog concentrates more than the null would predict.

**Training protocol and metric.** We split the real catalog into train/test subsets; for each subset we generate matched null samples by shuffling distances within that subset, ensuring that evaluation uses disjoint data. We then train a small MLP on the combined (real + null) training set and

| Group | $N$ | bw (Mpc) | KDE test LL | NCE AUC |
|---|---|---|---|---|
| SNe (all) | 11514 | 50 | -18.887 | 0.523 |
| GRBs (all) | 522 | 100 | -22.824 | 0.516 |
| SNe: Ia | 8262 | 35 | -18.944 | 0.535 |
| SNe: II | 1912 | 75 | -17.875 | 0.515 |
| SNe: Ibc | 552 | 50 | -17.299 | 0.551 |
| SNe: IIn | 301 | — | — | — |
| SNe: Other/Unknown | 487 | — | — | — |
| GRBs: long | 485 | — | — | — |
| GRBs: short | 37 | — | — | — |

Table 3. Density and density-ratio baselines in comoving 3D. KDE: Gaussian kernel, CV bandwidth. NCE: MLP vs. selection-preserving null; test AUC.

report the ROC-AUC on the held-out test set. Values near 0.5 indicate weak distinguishability (the catalog is close to a selection-preserving null under the chosen representation), while values substantially above 0.5 indicate detectable structure beyond marginal sky and distance distributions.

**Interpretation.** Crucially, this evaluation is not a claim of astrophysical clustering in an absolute sense. It is a test of *detectable 3D dependence between angular position and distance* after controlling for the strongest observational marginals. This makes the baseline conservative and particularly suitable for heterogeneous survey products.

## 6.3. Results

Table **??** reports KDE held-out log-likelihood and NCE ROC-AUC for the full SNe and GRB point clouds, as well as for common subcategories when available. Because the null preserves angular and radial marginals, improvements over chance in the NCE test indicate structure that is not explained solely by footprint and distance selection.

## 7. Figures and Qualitative Results

The pipeline produces a large set of figures under `paper_figures/`. In this workshop paper we include a representative subset.

## 8. Geometric Covers for Catalog Summarization

Beyond global scores, we aim to produce *compact, interpretable summaries* of each 3D catalog that can be visualized, compared across categories, and used for selection of representative follow-up targets. We therefore summarize each point cloud by a small set of representative centers together with error radii that quantify how well the centers cover the distribution.

**Definitions.** Let $\mathcal{P} = \{\mathbf{p}_1, \ldots, \mathbf{p}_N\}$ be the set of points in $\mathbb{R}^3$. A cover is specified by $K$ centers $\mathcal{C} = \{\mathbf{c}_1, \ldots, \mathbf{c}_K\}$.

For any point $\mathbf{p}$, define the point-to-center distance

$$r(\mathbf{p}) = \min_{\mathbf{c} \in \mathcal{C}} \|\mathbf{p} - \mathbf{c}\|. \tag{10}$$

We also define an induced assignment $a(\mathbf{p}) = \arg\min_{\mathbf{c} \in \mathcal{C}} \|\mathbf{p} - \mathbf{c}\|$ (ties broken arbitrarily), which partitions the catalog into $K$ subsets.

**Objectives and algorithms.** We consider two standard objectives that emphasize different notions of error:
- **K-center (minimax):** choose $\mathcal{C}$ to minimize the maximum assignment distance,

$$\min_{\mathcal{C}} \max_i r(\mathbf{p}_i), \tag{11}$$

approximated with farthest-first traversal [22]. This is well suited when the goal is a guaranteed worst-case coverage radius.
- **K-means (least squares):** choose $\mathcal{C}$ to minimize the sum of squared assignment distances,

$$\min_{\mathcal{C}} \sum_{i=1}^{N} r(\mathbf{p}_i)^2, \tag{12}$$

using standard K-means [38]. This is often more faithful to high-density regions but can allow larger worst-case errors in sparse tails.

**From centers to "balls": per-center radii.** For interpretability, we convert assignments into *per-center error radii*. For a center $\mathbf{c}_k$, let $\mathcal{P}_k = \{\mathbf{p} \in \mathcal{P} : a(\mathbf{p}) = k\}$ and define the set of within-cluster distances $\{\|\mathbf{p} - \mathbf{c}_k\| : \mathbf{p} \in \mathcal{P}_k\}$. We report multiple summaries: (i) the maximum radius (worst-case), (ii) a robust quantile radius (e.g., 95th percentile), and (iii) the mean/median radius. The quantile radii yield a set of $K$ balls that "mostly" cover the distribution, with outliers explicitly visible as points outside the balls.

**Why covers?** Covers provide a compact representation that is useful for: (i) visualizing large catalogs (a few centers vs. thousands of points), (ii) selecting representative follow-up targets or anchor points for downstream modeling, (iii) defining coarse partitions of space that can be compared across categories or surveys, and (iv) compressing catalogs for interactive exploration (especially when combined with PCA or $\log_{10} d_c$ reparameterizations). Because the cover construction is purely geometric, it can be applied uniformly across SNe, GRBs, and clusters.

**Diagnostics across $K$.** In addition to single-$K$ summaries, we typically compute covers for a sweep of $K$ (e.g., $K = 1, \ldots, 9$) and examine how errors decay. This

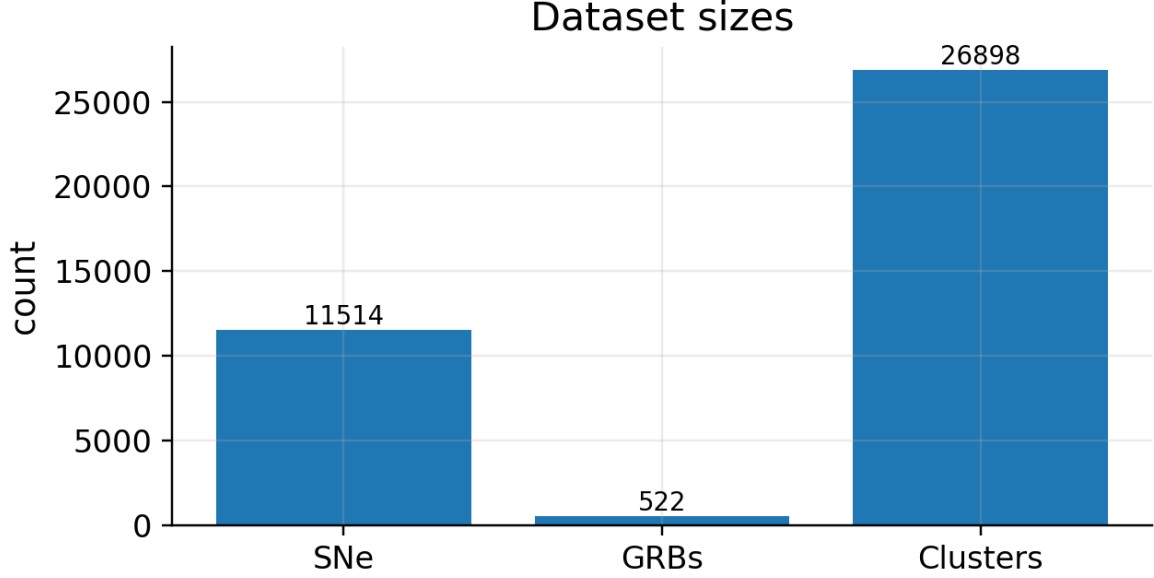

Figure 1. Dataset size overview. This figure is generated automatically from the processed tables.

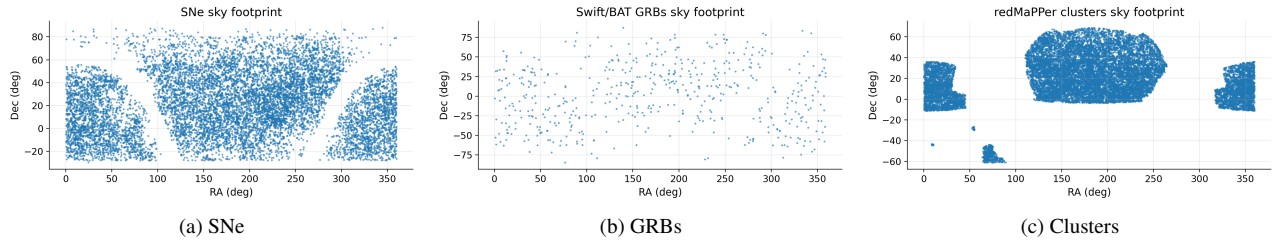

(a) SNe  (b) GRBs  (c) Clusters

Figure 2. Sky footprints (RA/Dec) of each catalog after preprocessing.

produces *coverage curves* such as max-radius vs. $K$ and quantile-radius vs. $K$, which provide an interpretable notion of "how many representatives are needed" to summarize the catalog at a desired error tolerance. These curves are especially informative when comparing different cover objectives (K-center vs. K-means) and different subcategories.

**Visualizations.** The accompanying scripts generate: (i) static plots such as PCA projections with Voronoi-like partitions, per-cluster hull/ellipse overlays, and distance-to-center distributions, and (ii) interactive HTML visualizations (drag/rotate 3D scatter with colored assignments and optional sphere overlays). We additionally export per-center tables to facilitate direct inclusion in the paper (center coordinates, radii, cluster sizes, and category composition). Examples and additional diagnostics are shown in Section 7.

## 9. Category Composition

Table **??** lists the category counts used in stratified analyses. For SNe, we use the reported/claimed type string (and optionally a normalized "major type" mapping) to define subcategories; for GRBs, we use duration-based classes (e.g., short vs. long) when available. We emphasize that the observed composition reflects both astrophysical rates and selection effects (e.g., classification completeness and redshift availability), so category imbalance should be considered when interpreting aggregate statistics or learning-based models.

**Imbalance and robustness.** When categories are highly imbalanced, global metrics can be dominated by the majority class. To mitigate this, we report both overall results and category-conditional diagnostics whenever sample sizes permit, and we treat rare categories either as "other" or as exploratory (visual-only) subsets. This is particularly im-

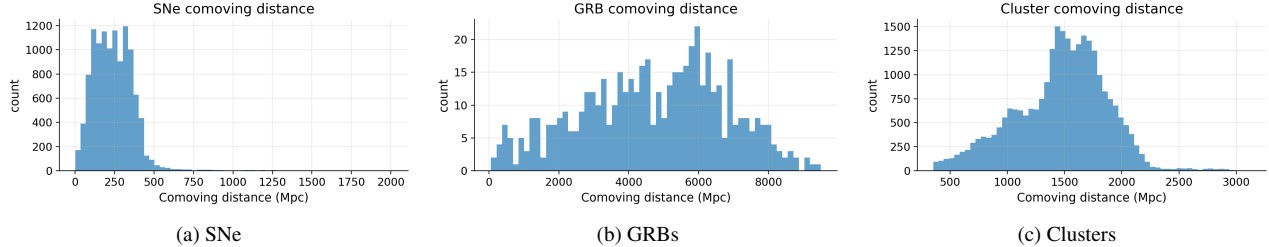

(a) SNe            (b) GRBs            (c) Clusters

Figure 3. Comoving distance distributions. SNe are concentrated at low redshift ($z \lesssim 0.52$), while GRBs extend to high redshift ($z$ up to $\approx 9.4$).

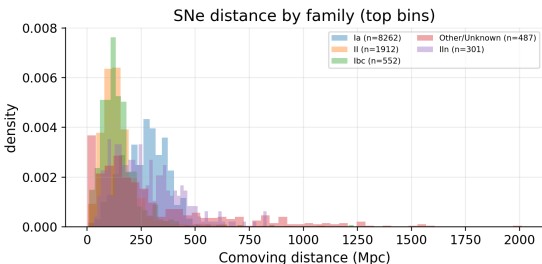

Figure 4. Distance distribution stratified by SN label.

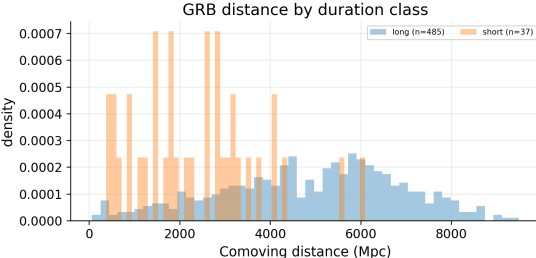

Figure 5. Distance distribution stratified by GRB duration class.

portant for cover-based visual summaries: sparse categories can appear as isolated tails that are poorly represented by centers optimized for the majority class.

## 10. Discussion and Limitations

The density-ratio AUCs in Table **??** are close to 0.5, suggesting that (at least for these coarse baselines and coordinate choices) the catalogs are only weakly distinguishable from the selection-preserving null. This is *expected*: survey selection and redshift incompleteness dominate many apparent trends, and astrophysical clustering in 3D is subtle for transients. Nevertheless, these baselines are valuable: they establish a quantitative floor for workshop submissions and highlight where additional modeling (e.g., hierarchical selection functions, host-galaxy information, or physically motivated priors) is needed.

| Category | $N$ | Fraction |
|---|---|---|
| **ZTF BTS SNe (claimed type)** | | |
| Ia | 8262 | 0.718 |
| II | 1912 | 0.166 |
| Ibc | 552 | 0.048 |
| Other/Unknown | 487 | 0.042 |
| IIn | 301 | 0.026 |
| **Swift/BAT GRBs (duration class)** | | |
| long | 485 | 0.929 |
| short | 37 | 0.071 |

Table 4. Basic category breakdown used for subset analysis and visualizations.

**What "error radii" mean.** Cover radii should be interpreted as geometric summary errors in comoving space, not as measurement uncertainties. They reflect how well $K$ centers can represent the spatial distribution under a chosen metric (Euclidean in comoving Mpc in this work).

**Missing or empty sources.** Some endpoints (notably the AstroCats quantity CSVs queried with `first&value`) may return placeholder responses depending on server state or throttling. The pipeline validates file contents, quarantines invalid downloads, and records missing-data provenance so that downstream analyses remain reproducible.

## 11. Conclusion

We introduce a geometry-first, fully reproducible pipeline that unifies public astronomical catalogs into a *3D comoving point-cloud*, logging preprocessing choices for deterministic regeneration. We add two conservative structure baselines: Gaussian KDE likelihoods and a classifier density-ratio test against a *selection-preserving null* that preserves footprint and radial profile but breaks 3D correlations. Catalogs are summarized via compact geometric covers (K-means/K-center) yielding representative events with error radii, plus tables and static/interactive visuals; future work adds uncertainty/host features, and richer selection.

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
