# OpenReview forum: "Cosmic Covers: A Geometry-First Pipeline for 3D Comoving Astronomical Catalogs"
_thecvf.com/CVPR/2026/Workshop/3D4S — CVPR 2026 Workshop 3D4S Poster_

### Official Review · Reviewer_k1i6 · 2026-04-20
**A carefully written infrastructure paper with a reproducible catalog pipeline, though a gap between claimed contributions and reported results weakens the contribution**

**Rating:** 6
**Confidence:** 2

**Review:**

Cosmic Covers presents a reproducible pipeline for harmonizing public astronomical catalogs (ZTF BTS supernovae, Swift/BAT GRBs, redMaPPer galaxy clusters) into unified 3D comoving point clouds, then summarizing them with KDE density baselines, noise-contrastive density-ratio tests against a selection-preserving null, and geometric covers (K-means and K-center with error radii). The citation list is extensive (67 references) and almost entirely clean. However, there is a notable gap between the claimed contributions and what is actually demonstrated in the paper. The geometric covers have no quantitative results reported: no coverage radii, no coverage curves, no summary tables. The density baselines are computed for 5 of 9 subgroups, and those that are computed yield NCE AUCs that are barely above chance, which the paper honestly acknowledges. Additionally, every in-text table reference is broken ("Table??" appears). The pipeline design is thoughtful, the mathematical exposition is clear, and the workshop fit is strong.

---

### Official Review · Reviewer_m4LX · 2026-04-25
**This paper presents a geometry-first pipeline that converts heterogeneous astronomical catalogs (SNe, GRBs, clusters) into a unified 3D comoving point cloud representation, followed by simple geometric analysis such as KDE-based density estimation, density-ratio testing against a selection-preserving null, and K-means/K-center covers for summarization. The focus is on building a reproducible baseline pipelin rather than proposing a new learning method.**

**Rating:** 6
**Confidence:** 3

**Review:**

### **Pros**

- **Strong emphasis on reproducibility and clean pipeline design.**
  The paper is very systematic in data ingestion, preprocessing, and coordinate transformation, making the pipeline easy to follow and potentially useful as a baseline.

- **Clear geometric formulation.**
  The conversion from $(\alpha, \delta, z)$ to 3D coordinates and the use of simple geometric tools (e.g., covers, KDE) are well explained and intuitive.

---

### **Cons**

- **Very limited novelty (main issue).**
  Most components are standard: coordinate conversion, KDE, NCE, K-means/K-center. The paper explicitly states it is a “baseline pipeline,” and there is no new algorithmic or modeling contribution.

- **Weak empirical signal / results are not compelling.**
  The key result shows AUC values close to 0.5, meaning the method barely distinguishes real data from the null model. This suggests the approach does not extract meaningful structure. (See discussion on page 8.)

- **Questionable fit for CVPR / 3D vision.**
  The work is closer to data processing and scientific pipeline construction than computer vision. There is no use of modern 3D learning methods, representations, or vision-specific insights.

---

### Decision · Program_Chairs · 2026-04-28

Accept (Poster)